# Overview of Radiological Studies on Visualization of Gubernaculum Tracts of Permanent Teeth

**DOI:** 10.3390/jcm10143051

**Published:** 2021-07-09

**Authors:** Masafumi Oda, Ikuko Nishida, Manabu Habu, Osamu Takahashi, Hiroki Tsurushima, Taishi Otani, Daigo Yoshiga, Katsura Saeki, Tatsurou Tanaka, Nao Wakasugi-Sato, Shinobu Matsumoto-Takeda, Yutaro Nagasaki, Ikuya Miyamoto, Shinji Kito, Masaaki Sasaguri, Yasuhiro Morimoto

**Affiliations:** 1Division of Oral and Maxillofacial Radiology, Kyushu Dental University, Kitakyushu 803-8580, Japan; r07oda@fa.kyu-dent.ac.jp (M.O.); t-tanaka@kyu-dent.ac.jp (T.T.); r16wakasugi@fa.kyu-dent.ac.jp (N.W.-S.); s-takeda@kyu-dent.ac.jp (S.M.-T.); r20nagasaki@fa.kyu-dent.ac.jp (Y.N.); 2Division of Developmental Stomatognathic Function Science, Kyushu Dental University, Kitakyushu 803-8580, Japan; nishida@kyu-dent.ac.jp (I.N.); katsura@kyu-dent.ac.jp (K.S.); 3Division of Maxillofacial Surgery, Kyushu Dental University, Kitakyushu 803-8580, Japan; h-manabu@kyu-dent.ac.jp (M.H.); r07takahashi@fa.kyu-dent.ac.jp (O.T.); r13sasaguri@fa.kyu-dent.ac.jp (M.S.); 4Division of Oral Medicine, Kyushu Dental University, Kitakyushu 803-8580, Japan; r17tsurushima@fa.kyu-dent.ac.jp (H.T.); r17otani@fa.kyu-dent.ac.jp (T.O.); r11yoshiga@fa.kyu-dent.ac.jp (D.Y.); 5Division of Oral and Maxillofacial Surgery, Iwate Medical University, Iwate 028-3694, Japan; ikuyam@iwate-med.ac.jp; 6Division of Dental Radiology, Meikai University School of Dentistry, Saitama 350-0283, Japan; kito@dent.meikai.ac.jp

**Keywords:** gubernaculum tract, successional tooth, accessional tooth, imaging, CT, odontoma, odontogenic tumor, odontogenic cyst

## Abstract

The eruption pathway from the dental follicle to the gingiva for permanent teeth is known as the gubernaculum tract (GT), a physiologic structure thought to play a role in tooth eruption. Cone beam computed tomography and multi-detector computed tomography have recently been used to visualize the GT, with the results indicating that this structure might be related to the normal eruption of teeth. By contrast, curved and/or constricted GTs may lead to abnormal tooth eruption. In addition, complex odontomas have been reported from within the GT or dental sac of unerupted permanent teeth. If an odontoma occurs within the GT, the tooth will not erupt normally. Moreover, the imaging characteristics of the GT from the top of the odontogenic mass to the alveolar crest are extremely useful for making a differential pathological diagnosis and for differentiating between odontogenic and non-odontogenic masses. Therefore, radiological studies on the GT have been attracting increasing attention. Given this background, the present review aims to clarify the imaging characteristics and review recent studies on the GT considering the importance of the research.

## 1. Introduction

Successional permanent teeth erupt from the same places where the primary teeth are shed; those without a deciduous predecessor grow in the distal region of the dental arch. The eruption pathway for both types of permanent teeth that runs from the dental follicle to the gingiva is called the gubernaculum dentis (GD) [1,2,3]. Histologically, the GD is composed of a fibrous band called the gubernacular cord (GCo), which typically consists of the epithelial remnants of the dental lamina, as well as a bony channel around the GCo connecting the pericoronal follicular tissue of the successional or accessional tooth with the overlying gingiva that opens to the alveolar bone crest [1,2,3]; this bony channel is known as the gubernaculum tract (GT) or gubernacular canal. However, few clinical studies have focused on the presence, appearance, and importance of these structures, which could be related to the normal eruption of teeth [4,5,6]. One possible explanation for the lack of studies is that the GT is difficult to visualize on two-dimensional radiographs, such as dental and panoramic radiographs, because it is very thin (1–3 mm in diameter) and therefore often superimposed on images of deciduous teeth and trabeculae [4,5,6]. Therefore, the GD has not received sufficient attention in dental fields such as pediatric dentistry and oral and maxillofacial radiology.

Due to their ability to obtain high-quality images of hard tissue, cone beam computed tomography (CBCT) and multi-detector computed tomography (MDCT) are considered among the most useful and reliable imaging modalities for examining the maxilla and mandible, including the teeth [7,8,9,10]. Accordingly, CBCT and MDCT are used extensively in dental fields, and their clinical application to numerous kinds of tooth- and alveolar bone-related diseases has rapidly advanced [7,8,9,10,11]. Our previous study was the first to describe and analyze the imaging characteristics of the GT [12]. In addition, our research group has conducted detailed analyses of the relationship between abnormal tooth eruption and curved/constricted GTs and major odontomas occurring in the GT or dental sac (DS) [13,14,15,16]. We reported that the imaging characteristics of GTs at the top of odontogenic and non-odontogenic masses were useful for differential diagnoses [13,14,15,16]. Subsequently, radiological studies on GTs, including our previous reports, began receiving more attention [5,6,17,18,19,20]. In this review, we summarize our previous studies, including those on the imaging characteristics of GTs, and review the recent literature on GTs.

## 2. Visualization of GT in Permanent Tooth on CT

### 2.1. CT Images of the GT in the Successional Tooth

As the CT image of the representative successional teeth, the GT of a maxillary central incisor with normal eruption is shown in Figure 1 [12]. As shown in Figure 1A, the GTs in the maxillary central and lateral incisors and canines are visualized on an axial CT image as round-shaped areas with low density on the palatal side of the deciduous predecessor. The diameters of the GTs are all less than 3 mm. As shown in Figure 1B, each GT is observed as a low-density corticated tract contiguous with the dental follicle of the unerupted maxillary central incisor on a sagittal CT image. Figure 1C shows that the GT becomes shorter as the central incisor erupts. As shown in Figure 1D, the GT disappears at the point when the DS of the unerupted central incisor becomes contiguous to the alveolar crest [12]. The GTs in the maxillary lateral incisors, canines, and premolars, and mandibular central and lateral incisors, canines, and premolars also show the same appearance on the CT. The detection rate for GTs in the maxillary anterior teeth and mandibular anterior teeth and premolars is approximately 95% on the CT [12]. The detection rate for GTs in the maxillary premolars is approximately 50% because in these teeth, the GT runs near the periodontal space of the predecessor via the bifurcation [12].

### 2.2. CT Images of GT in Accessional Tooth

The GTs in the maxillary and mandibular molars as accessional teeth are known to be larger than those in the anterior teeth [12,16]. Other GT peculiarities can be observed in the molar region, including the alteration of the GT form of the maxillary and mandibular third molars (Figure 2 and Figure 3) [16]. In the early stage, the shape of the GT is a that of a groove on the mesial alveolar crest that runs continuous with the DS to the mesial tooth bud (Figure 2A and Figure 3A). Next, as the alveolar bone around the GT develops, the GTs in the third molar form a J-shape and a Y-shape in the maxillary and mandibular teeth, respectively (Figure 2B and Figure 3B). Then, in the mature stage, the course of the GT becomes perpendicular to the alveolar crest (Figure 2C and Figure 3C). Some GT forms have also been identified in the first and second molars. The primary difference in comparison with successional teeth is that in accessional teeth, the dental lamina is derived from the mesial tooth as opposed to the predecessors. This peculiarity, the way in which the GT in the molar is not perpendicular to the alveolar crest before the mature stage, reflects embryological features. Another notable peculiarity in the mandibular molar region is the “pseudo-GT” that extends from the distal side of the mandibular third molars (Figure 3D) [16]. The pseudo-GT tends to disappear with age without generating a fourth molar. Taken together, these molar GT morphologies closely resemble the three-dimensional figures of the dental lamina often shown in oral histology textbooks [21].

### 2.3. CT Images of GT in Maxillary Anterior Teeth with Delayed Eruption and Mesiodens

As the CT image of the representative maxillary anterior teeth with delayed eruption, the GT of maxillary central incisor with delayed eruption is shown in Figure 4 [12,13]. The GT in the maxillary central incisor with delayed eruption appears very similar to that with normal eruption on CT, but the tract to the tooth axis angle is greater than that for normal eruption (Figure 4). Therefore, a significant difference in angulation has been shown to be one of the signs of delayed eruption [13].

As the CT image of the representative impacted supernumerary teeth, the GT of mesiodens is shown in Figure 5 [13]. The GT in the inverted mesiodens is derived from the incisive canal (Figure 5A), whereas that in the mesiodens in the normal direction is derived from the alveolar crest (Figure 5B). In inverted mesiodens, the connecting area of the major GTs is the cervical or root area (Figure 5A), whereas that in mesiodens with normal eruption is the crown area (Figure 5B). In addition, it has been reported that the detection rate for GTs is significantly lower in impacted mesiodens and anterior teeth with delayed eruption than in teeth with normal eruption [13].

Various causes of tooth eruption disturbance have been reported [22,23,24]. Before visualizing GTs on CT, the major causes were considered to be physical obstructions, such as the presence of supernumerary and/or impacted teeth and space-occupied lesions [22]. However, following visualization of the GTs in teeth by our research group, the causes of tooth eruption disturbance were considered more complicated than physical obstructions, as the involvement of the GT became clear [12,13,14,15]. Compared with teeth with normal eruption, the GTs in teeth with delayed eruption have also been shown to have higher angulation of the tract to tooth axis, as described above (Figure 4) [13]. This finding made it clear that curvature and/or constriction of the GT could also be a reason for tooth eruption disorders [12,13]. Furthermore, some odontogenic lesions have been shown to occur in the GTs in permanent teeth [14,15]. We consider this in the next section.

## 3. Relationship between GT and Odontomas

Our previous investigation revealed significant insights into the occurrence of odontomas [14]. We reported that all odontomas could be divided into the four groups based on the spatial association between odontomas and GTs or the DS (Figure 6). In that study the groups were as follows: Group 1 (Figure 6A) consisted of odontomas within the GT in the unerupted permanent successor; Group 2 (Figure 6B) consisted of odontomas within the DS of the unerupted permanent successor; Group 3 (Figure 6C) consisted of odontomas that were not detected in either the GT or the DS of the unerupted permanent successor, but the GT was present; and Group 4 (Figure 6D) consisted of odontomas unrelated to the GT. The vast majority of the odontomas belonged to Groups 1–3, with only about 12% belonging to Group 4 [14].

The first significant indication was that odontomas occur in the GTs or DS of unerupted permanent teeth (Groups 1 and 2) [14]. The results of our study revealed that odontomas could directly hinder tooth eruption from inside the GT, which is one of the critical causes of eruption disturbance along with the geometrics described in the previous paragraph [14]. The second important insight was that 70% of odontomas without unerupted teeth were also connected to GTs (Group 3) [14]. In such cases, a hamartoma may occur as opposed to a tooth bud at the tip of the dental lamina.

At the same time, our results influenced the radiological findings of odontomas before and after visualizing GTs on CT. After our report, radiologists began to suggest that the inclusion of unerupted teeth in the GT or DS should be added as a characteristic finding of odontomas on CT. The presence or absence of the GT could be useful for differentiating odontomas, especially the complex type, from other hard tissue lesions such as fibrous dysplasia and cemento-ossifying fibroma [14]. These findings suggest that the radiological characteristics of GT in relation to other odontogenic lesions are very important. This is considered and reviewed in the next section.

## 4. Significance of GTs for Differential Diagnosis between Odontogenic and Non-Odontogenic Masses

As mentioned earlier, about 80% of odontomas have GTs, which indicates that the presence or absence of GTs could be useful for a differential diagnosis between complex odontomas and jaw lesions with high-density structures [14]. Therefore, in a subsequent study, we retrospectively investigated whether odontogenic and non-odontogenic masses, including tumors and cysts, significantly differed in regard to the detection level of GT [15]. The results indicated that, in contrast to non-odontogenic tumors or cysts, intact GTs or broad pathways were detected at the top of about 90% of odontogenic tumors or cysts on CT [15], suggesting that the broad pathway was an expanded GT. The presence or absence of GTs in masses could therefore be a very useful finding for the differential diagnosis between odontogenic and non-odontogenic tumors or cysts [15]. Defect areas were detected at the top of the alveolar bone that included an expanded GT in ameloblastomas (Figure 7A) or dentigerous cysts (Figure 7B) as representative odontogenic tumors or cysts. Conversely, defect areas at the top of the alveolar bone were not detected in ossifying fibromas (Figure 7C) or simple bone cysts (Figure 7D).

We also evaluated the association between the size of masses and alterations in GTs among amoloblastomas, dentigerous cysts, and odontogenic keratocysts as representative odontogenic masses [15]. The results revealed a characteristic altered GT in accordance with a larger mass. A strong significant correlation was found between the expansion of the GTs and tumor size in ameloblastomas, but only a weak correlation was found in dentigerous cysts, and no correlation was seen in odontogenic keratocysts [15]. These imaging characteristics of GTs at the top of masses are therefore useful for both differential and pathological diagnoses of odontogenic masses [15].

## 5. Recent Radiological Investigations and Future Studies on GTs

In our previous reports, we described the successful visualization of the GT in teeth on CT and elucidated the significance of GTs for normal tooth eruption, the characteristics of GTs in delayed or supernumerary teeth, and the usefulness of differential diagnoses between odontogenic and non-odontogenic masses [12,13,14,15,16]. Subsequently, numerous reports on the GT were published [5,6,17,18,19,20]. The GCo was previously considered not to be associated with the eruption process [3]. However, some studies on the GT and abnormal eruption were carried out after we reported our results [5,6]. Those studies reported that the detection rates of GTs in not only anterior teeth with delayed eruption, but also premolars and molars, were significantly lower than those with normal eruption, and that the GTs in teeth with disturbed eruption showed abnormalities. Therefore, the existence of GT abnormalities in teeth on CT may help predict normal or abnormal tooth eruption. In other words, if the GT is present in a tooth, normal eruption can be expected. However, if the GT is absent, it might indicate abnormal or delayed eruption or an impacted tooth [5,6,12,13]. If the GT is curved and/or constricted, abnormal tooth eruption such as dislocation or impaction may be likely [5,6,12,13]. High-resolution MRI can visualize GT as a radiation-free alternative in children. Further studies may elucidate on the obscure parts of these findings.

In another study, GTs in regional odontodysplasia teeth—the so-called “ghost teeth”—were identified, and all right mandibular teeth showed wide pulp chambers and shortened roots with open apices and thin dentinal walls [18]. Normal eruption was obscure in these odontodysplasia teeth, so a follow-up study is still required. In addition, we expect the characteristics of GTs in various teeth-related diseases to be reported in the near future. In particular, studies on the characteristics and dynamics of some teeth-related diseases that involve a large number of impacted teeth, such as cleidocranial dysplasia, could clarify the significance of the GT with regard to tooth eruption.

Another study investigated whether the presence or absence of the GT was related to odontogenic or non-odontogenic masses, including tumors and cysts [19]. Again, the imaging characteristics of GTs at the top of masses are expected to be very useful for this purpose [15,19,20]. If the GT at the top of the alveolar bone can be detected in masses, pathologically, it should be diagnosed as an odontogenic tumor or cyst; otherwise, it should be diagnosed as a non-odontogenic tumor or cyst. GTs have long been believed to be of the origins of odontogenic tumors, such as ameloblastomas in the 1960s [25] and adenomatoid odontogenic tumors in 1992 [26,27]. Taken together with our series of studies and that by Kamarthi et al. [19], we speculated, based on radiological studies, that the origins of odontogenic tumors and cysts may be found within the GT [15]. The epithelial remnants of the dental lamina within the GT can also be the origin of odontogenic lesions. We recently reported additional evidence for this in accessional teeth, as mentioned above [16]. Pseudo-GTs—tracts continuing from the distal side of the mandibular third molars—have been visualized in many cases involving children [16]. Basically, no teeth have been found at the distal side of the mandibular third molars; however, pseudo-GTs could be visualized, and these disappear with age. Odontogenic masses such as ameloblastomas and odontogenic keratocysts frequently occur on the distal side of the mandibular third molars. Therefore, the extension of GTs to the distal side of the mandibular third molars may be one of the origins of odontogenic masses such as tumors and cysts. This remains to be elucidated in future research.

## 6. Conclusions

CT images of GT indicates that this structure might be related to the normal eruption of teeth. Curved and/or constricted GTs on CT may lead to abnormal tooth eruption. On CT, odontomas are included within the GT or dental sac of unerupted permanent teeth. If an odontoma occurs within the GT, the tooth will not erupt normally. The imaging characteristics of the GT from the top of the odontogenic mass to the alveolar crest are extremely useful for making a differential pathological diagnosis and for differentiating between odontogenic and non-odontogenic masses. Therefore, radiological studies on the GT have been attracting increasing attention.

## Figures and Tables

**Figure 1 jcm-10-03051-f001:**
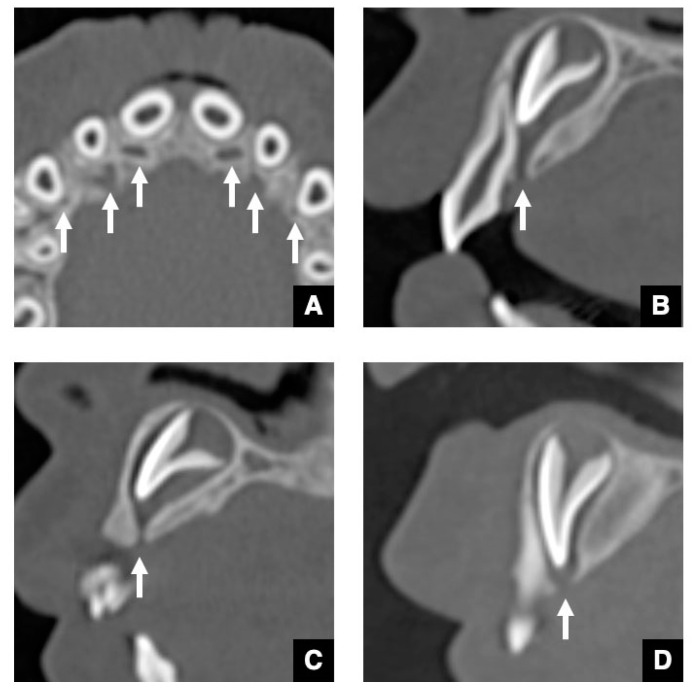
CBCT images of various periods of GTs in maxillary central incisors with normal eruption as representative successional teeth. (**A**) The GTs in anterior teeth (arrows) appear as six round-shaped areas with low density on the palatal side of the deciduous predecessor on an axial CBCT image. (**B**) The GT (arrow) was viewed as a low-density corticated tract contiguous with the dental follicle of an unerupted maxillary central incisor on a sagittal image. (**C**) The GT (arrow) becomes shorter as the unerupted central incisor erupts in the alveolar crest on a sagittal image. (**D**) The GT disappears (arrow) when the DS of the unerupted central incisor becomes contiguous to the alveolar crest on a sagittal image.

**Figure 2 jcm-10-03051-f002:**
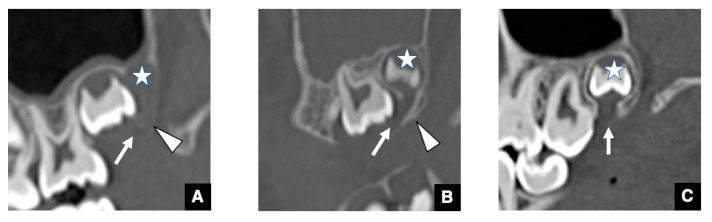
Representative CBCT images of various periods of GTs in maxillary molar regions with normal tooth eruption. (**A**) GT (arrow) of the maxillary third molar bud (star) in the stage of the groove-shaped GT on a parallel sectional CBCT image. The alveolar crest bone (arrowhead) on the GT is not observed because of immaturity. (**B**) GT (arrow) of the maxillary third molar (star) in the J-shape stage on a parallel sectional CBCT image. The alveolar crest bone (arrowhead) on the GT appears, and the ductal form becomes clear. Notably, the GT curves mesial and opens to the mesial tooth bud. As a result, the GT forms a J-shape. (**C**) GT (arrow) of the maxillary third molar (star) in the mature stage on a parallel sectional CBCT image. The GT runs perpendicularly from the DS to the alveolar crest.

**Figure 3 jcm-10-03051-f003:**
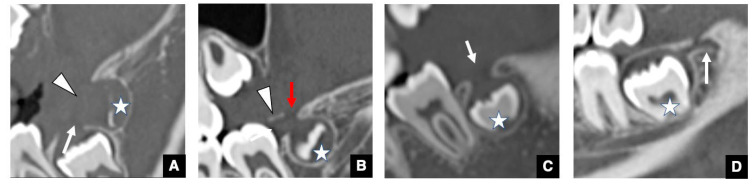
Representative CBCT images of various periods of GTs in mandibular molar regions with normal tooth eruption, (**A**) GT (arrow) of the maxillary third molar bud (star) in the stage of groove-shaped GT on a parallel sectional CBCT image. The alveolar crest bone (arrowhead) on the GT is not observed because of immaturity. (**B**) GT (arrow) of the mandibular third molar (star) in the Y-shape staged on a parallel sectional CBCT image. The alveolar crest bone (arrowhead) on the GT appears, and the ductal form becomes clear. Notably, the GT consisted of two forms in the penetrating area: one was directed mesially as a parallel part (arrow), and the other was directed vertically to the alveolar crest as a perpendicular part (red arrow). As a result, the GT forms a Y-shape. (**C**) GT (arrow) of the mandibular third molar (star) in the mature stage on a parallel sectional CBCT image. The GT runs perpendicularly from the DS to the alveolar crest. (**D**) CBCT image of the pseudo-GT (arrow) continuing from the distal side of the mandibular third molar (star).

**Figure 4 jcm-10-03051-f004:**
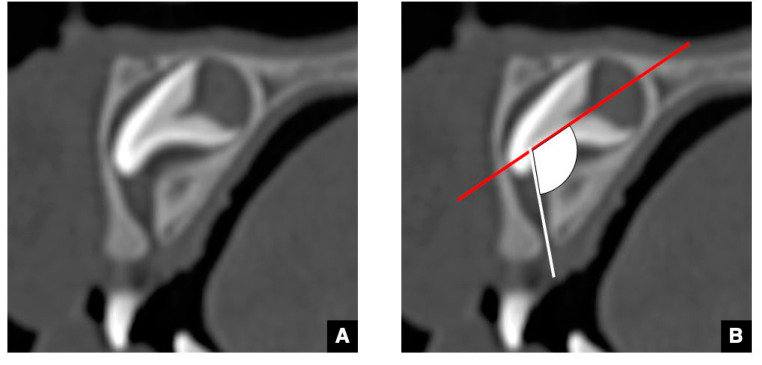
CBCT image (**A**) and illustration (**B**) of GT in the maxillary central incisor with delayed eruption. GT (arrow) of the maxillary central incisor with delayed eruption appears very similar to that with normal eruption on CBCT, but the angulation (fan) of the tract (line) to tooth axis (red line) tends to be greater than that with normal eruption on the sagittal image shown in Figure 1.

**Figure 5 jcm-10-03051-f005:**
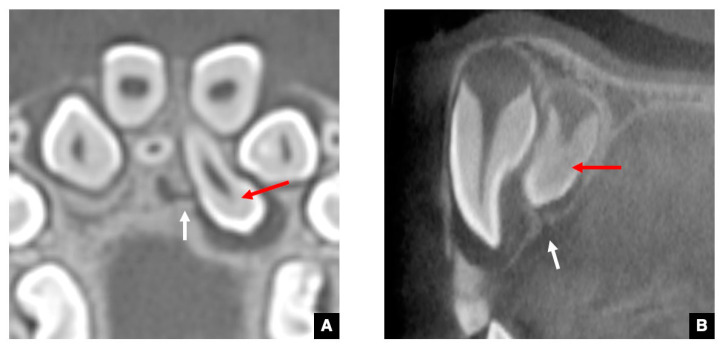
CBCT images of GTs in mesiodens. (**A**) The GT (arrow) of the inverted mesiodens (red arrow) is derived from the incisive canal on a CBCT axial image. The major connecting area of the GTs in inverted mesiodens is the cervical area. (**B**) The GT (arrow) of the mesiodens (red arrow) with normal eruption is derived from the alveolar crest on a CBCT sagittal image. The connecting area of the major GTs in the mesiodens with normal eruption is the crown area.

**Figure 6 jcm-10-03051-f006:**
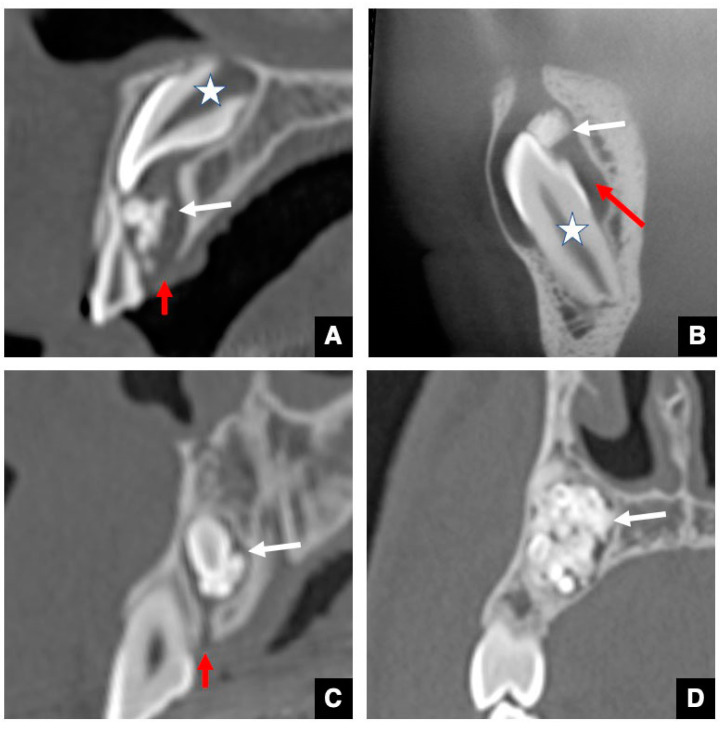
Typical spatial relationship between odontomas and GT or DS on CT. (**A**) CBCT image of the odontoma classified as Group 1. Odontoma (arrow) is detected within the GT (red arrow) of the unerupted permanent successor (star). (**B**) CBCT image of the odontoma classified as Group 2. Odontoma (arrow) is detected within the DS (red arrow) of the unerupted permanent successor (star). (**C**) CBCT image of the odontoma classified as Group 3. Odontoma (arrow) is not detected in either the GT or the DS of the unerupted permanent successor, but its own GT (red arrow) is present. (**D**) CBCT image of the odontoma classified as Group 4. The odontoma (arrow) did not have a GT.

**Figure 7 jcm-10-03051-f007:**
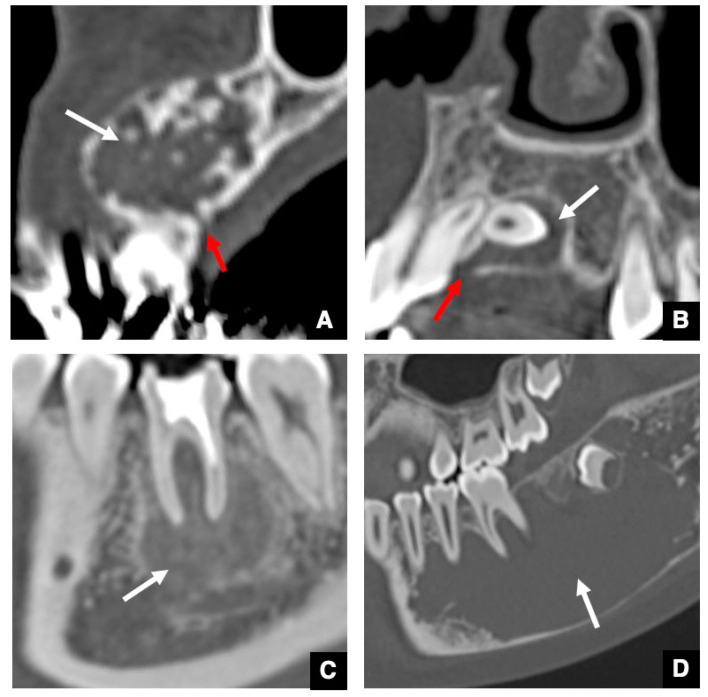
Difference in GTs between odontogenic and non-odontogenic masses. (**A**) The GT (red arrow) continues to the top of the ameloblastoma (arrow) as the representative odontogenic tumor. (**B**) An almost intact GT (red arrow) continues to the top of the dentigerous cyst (arrow) as the representative odontogenic cyst. (**C**) The GT at the top of the alveolar bone is not detected in an ossifying fibroma (arrow) as the representative non-odontogenic tumor. (**D**) The GT at the top of the alveolar bone is not detected in a simple bone cyst (arrow) as the representative non-odontogenic cyst.

## Data Availability

Not applicable.

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
