# Peer review of "Overview of Radiological Studies on Visualization of Gubernaculum Tracts of Permanent Teeth"

_jcm, 2021, doi:10.3390/jcm10143051_

Round 1

Reviewer 1 Report

Very nice investigation. I was wondering in which context the CBCTs were created? Was there an indication to use the radiation on children? 

Is there any data on how high the radiation exposure was? 

Do the authors think that the gubernaculum tract could also be visualised by high-resolution MRI as a radiation-free alternative in children? Please comment on this. 

Author Response

  1. The present review was retrospectively conducted using CT or CBCT scans that were obtained from children with oral and maxillofacial diseases.
  2. The exposure of CT or CBCT was appropriate to evaluate the children with oral and maxillofacial diseases.
  3. We added the sentence “High-resolution MRI can visualize GT as a radiation-free alternative in children. Further studies may elucidate to be obscure.” in line 258.

Reviewer 2 Report

The article is very interesting from a scientific point of view, unfortunately there is no summary that highlights the importance of the research.

Figure descriptions should appear below them.

I think there is an error in line 176 in the description of Group 3.

Author Response

  1. The importance of the research was added in last sentence of Abstract.
  2. Figure descriptions were appeared below them.
  3. We revised the description of Group 3 in line 180.

Reviewer 3 Report

The article entitled " Overview of radiological studies on visualization of gubernaculum tracts of permanent teeth"  investigated the gubernaculum tracts of permanent teeth. This topic is very interesting for the scientific community. The sections introduction, material and methods, and discussion are well described,  The authors may add the conclusion section to synthesize the research giving a clinical approach and suggestions.

Author Response

We add the conclusion section to synthesize the research giving a clinical approach and suggestions.

Reviewer 4 Report

The topic treated by the authors is certainly interesting and also rarely treated in the literature. The authors use an “unconventional” study design ie they do a narrative review, but actually summarize the studies they have previously conducted. In the paragraph “Visualization of GT in permanent tooth on CT”, and several times in the text, the authors begin the paragraph with “Figure .. shows…”. I think this style needs to be changed. First a concept must be affirmed and then it must be confirmed by an image. 

Author Response

We revised the sentence “Figure .. shows…” in the paragraph “Visualization of GT in permanent tooth on CT”.

Round 2

Reviewer 4 Report

The requested changes have been made.

The manuscript is now publishable.